# Silver(I) Complexes Based on Oxadiazole-Functionalized *α*-Aminophosphonate: Synthesis, Structural Study, and Biological Activities

**DOI:** 10.3390/molecules27238131

**Published:** 2022-11-22

**Authors:** Shaima Hkiri, Kübra Açıkalın Coşkun, Elvan Üstün, Ali Samarat, Yusuf Tutar, Neslihan Şahin, David Sémeril

**Affiliations:** 1Synthèse Organométallique et Catalyse, UMR-CNRS 7177, University of Strasbourg, 4 rue Blaise Pascal, 67008 Strasbourg, France; 2Laboratory of Hetero-Organic Compounds and Nanostructured Materials (LR18ES11), Faculty of Sciences of Bizerte, University of Carthage, Bizerte 7021, Tunisia; 3Department of Medical Biology and Genetics, Faculty of Medicine, University of İstanbul Aydın, Istanbul 34295, Turkey; 4Department of Chemistry, Faculty of Art and Science, University of Ordu, Ordu 52200, Turkey; 5Department of Basic Pharmaceutical Sciences, Faculty of Pharmacy, University of Health Sciences-Turkey, Istanbul 34668, Turkey; 6Department of Science Education, Faculty of Education, University of Cumhuriyet, Sivas 58140, Turkey

**Keywords:** 1,3,4-oxadiazole, silver(I) complex, X-ray structure, in vitro cytotoxicity, MCF-7 cell line, PANC-1 cell line, molecular docking

## Abstract

Two silver(I) complexes, bis{diethyl[(5-phenyl-1,3,4-oxadiazol-2-yl-κ*N*^3^:κ*N*^4^-amino) (4-trifluoromethylphenyl)methyl]phosphonate-(tetrafluoroborato-κ*F*)}-di-silver(I) and tetrakis-{diethyl[(5-phenyl-1,3,4-oxadiazol-2-yl-κ*N*^3^-amino)(4-trifluoromethylphenyl)methyl]phosphonate} silver(I) tetrafluoroborate, were prepared starting from the diethyl[(5-phenyl-1,3,4-oxadiazol-2-yl-amino)(4-trifluoromethylphenyl)methyl]phosphonate (**1**) ligand and AgBF_4_ salt in Ag/ligand ratios of 1/1 and 1/4, respectively. The structure, stoichiometry, and geometry of the silver complexes were fully characterized by elemental analyses, infrared, single-crystal X-ray diffraction studies, multinuclear NMR, and mass spectroscopies. The binuclear complex ([Ag_2_(**1**)_2_(BF_4_)_2_]; **2**) crystallizes in the monoclinic asymmetric space group *P*2_1_/c and contains two silver atoms adopting a {AgN_2_F} planar trigonal geometry, which are simultaneously bridged by two oxadiazole rings of two ligands, while the mononuclear complex ([Ag(**1**)_4_]BF_4_; **3**) crystallizes in the non-usual cubic space group *Fd-3*c in which the silver atom binds to four distinct electronically enriched nitrogen atoms of the oxadiazole ring, in a slightly distorted {AgN_4_} tetrahedral geometry. The *α*-aminophosphonate and the monomeric silver complex were evaluated in vitro against MCF-7 and PANC-1 cell lines. The silver complex is promising as a drug candidate for breast cancer and the pancreatic duct with half-maximal inhibitory concentration (IC_50_) values of 8.3 ± 1.0 and 14.4 ± 0.6 μM, respectively. Additionally, the interactions of the ligand and the mononuclear complex with Vascular Endothelial Growth Factor Receptor-2 and DNA were evaluated by molecular docking methods.

## 1. Introduction

The coordination chemistry of silver(I), which displays a [Kr] 4d^10^ 5s^1^ electronic configuration, is particularly rich and can adopt a wide range of geometries [1,2,3]. In addition to linear silver(I) complexes [4,5,6], it is common to find triangular [7,8] or T-shaped [4,9], tetrahedral [10,11], or square plane [4,12] geometries but also compounds in which the silver atom is coordinated to five [13], six [13], seven [14], and even eight [15] ligands, often generating coordination polymers [16,17].

One of the most widely used coordinating atoms to silver has been found to be the nitrogen atom, which is generally incorporated in polydentate heterocyclic structures. Among these aza-heterocyclic ligands, 1,3,4-oxadiazoles have been well studied, leading to a wide coordination diversity of the silver cation [18,19,20,21,22,23,24,25,26,27,28]. For instance, Dong et al. reported the use of the 2,5-di(pyridin-4-yl)-1,3,4-oxadiazole ligand (**A**; Figure 1) with the AgSO_3_CF_3_ metal precursor (metal/ligand ratio: 1/2) in a CH_2_Cl_2_/MeOH mixture, to obtain the infinite one-dimensional polymeric compound {[Ag_4_(**A**)_4_](SO_3_CF_3_)_4_}_n_ in a 71% yield. The single-crystal analysis revealed the presence of two different Ag(I) centers: the first one adopted nearly a linear coordination with the two pyridyl N-atoms from two **A** ligands, while the second one had a distorted trigonal coordination environment made of three N-donor atoms of two pyridyl and the 1,3,4-oxadiazole moieties. The tetramer formed a 40-membered macrocycle, which was further connected to a smaller 14-membered dimeric unit [29]. The same group reported the reaction of 2-(5-(2-hydroxyphenyl)-1,3,4-oxadiazol-2-yl)phenyl isonicotinate (**B**; Figure 1) with an equimolar amount of AgPF_6_, which led to the formation of a bimetallic metallocycle of formula [Ag_2_(**B**)_2_·(MeOH)_2_](PF_6_)_2_ in which each Ag(I) center adopted a three-coordination sphere formed by one (*N*,*O*)-chelating donor and one pyridyl N-donor from a second **B** ligand. The two ligands arranged in a head-to-head fashion to form a distorted rectangular binuclear ring [30]. Recently, Qin et al. described the formation of a three-dimensional polymer from the 1,3,5-tris(5-(methylthio)-1,3,4-oxadia-zol-2-yl)benzene ligand (**C**; Figure 1) and AgSO_3_CF_3_ precursor (metal/ligand ratio: 3/1). The single crystal analysis revealed that the asymmetric unit is composed of two crystallographically independent silver cations, one ligand **C**, one coordinated water molecule, one coordinated triflate anion, a free counter anion, and a molecule of dichloromethane. The first Ag(I) adopts a distorted tetrahedral coordination sphere generated by four N(oxadiazole) atoms from four separate ligands. The second Ag(I) is also four-coordinated by two N(oxadiazole) atoms from two separate ligands and two oxygen atoms from a water molecule and a triflate anion. The polymeric network is generated by ligand **C**, which acts as a hexa-dentate ligand, binding to six Ag(I) cations. It is worth noting that, in the formed polymer, the sulfur atoms are not involved in the coordination with the metal [31]. An inorganic ribbon, formed by mixing (*E*)-3-(5-(furan-2-yl)-1,3,4-oxadiazol-2-yl)acrylic acid (**D**; Figure 1) and AgNO_3_ (metal/ligand ratio: 1/1) in the presence of a base, was reported by the group of Kokunov. In this polymer, two bidentate carboxylate groups bind two silver atoms and form dimeric blocks in which the Ag-Ag distance is only 2.854(1) Å. The distorted tetrahedral coordination of the Ag(I) cation is completed with an oxadiazole ring [32].

The possibility to modulate the coordination sphere of the silver(I) cation has allowed, in particular, the development of efficient silver complexes for medicinal applications, notably as antibacterial, antifungal, or anticancer agents [33,34,35,36,37,38,39,40,41]. In recent years, there has been growing interest in organic silver complexes as promising anticancer agents with efficacious activity against various cancer cell lines [42,43,44]. These complexes, which exhibit low toxicity for humans [45] and have long been used as antimicrobial drugs, are potential candidates to replace platinum-based anticancer drugs such as cisplatin (*cis*-diaminodichloroplatine(II) [Pt(NH_3_)_2_Cl_2_]) [46], which, despite their success, have drug resistance, undesirable side-effects such as gastrointestinal disorders, bleeding, allergic reactions, or a decrease in immunity, and are effective against only a few types of cancer [47].

Angiogenesis is a physiological process of formation and development of new vessels from existing vessels [48]. The growth and spread of solid tumor cells require tumor angiogenesis [49]. There are many growth factors involved in tumor angiogenesis, one of the most notable of these is the vascular endothelial growth factor (VEGF) family [50]. The vascular endothelial growth factor receptors (VEGFRs) in mammalian cells include three members: VEGFR-1, VEGFR-2, and VEGFR-3 [51]. Among these three receptors, VEGFR-2 is considered to be the most important marker for endothelial cell development and directly regulates tumor angiogenesis [52]. Some studies reported that VEGFR-2 has been found in colon cancer, pancreatic cancer, small cell lung cancer, and breast cancer cells [53]. Therefore, managing the action mechanism of VEGFR-2 could be an effective way in the treatment of several types of cancer.

In this context, we now report the coordination of diethyl[(5-phenyl-1,3,4-oxadia-zol-2-ylamino)(4-trifluoromethylphenyl)methyl]phosphonate to the silver cation AgBF_4_. The cytotoxicities of the *α*-aminophosphonate and one of its silver complexes were investigated toward the MCF-7 human breast cancer, which is the most frequently diagnosed cancer and the most common cause of death in women, and PANC-1 human pancreatic cancer cell lines. Our interest in this kind of ligand is stimulated by the well-known anticancer properties of 1,3,4-oxadiazoles [54] and α-aminophosphonates [55,56,57,58], which could be beneficial for obtaining novel silver(I) complexes with enhanced anticancer activity. Additionally, the interactions of *α*-aminophosphonate and one of its silver complexes against VEGFR-2 and DNA crystal structures were analyzed by molecular docking methods.

## 2. Results and Discussion

### 2.1. Silver(I) Complexes Synthesis

The slow diffusion of hexane into a THF solution of diethyl[(5-phenyl-1,3,4-oxadia-zol-2-ylamino)(4-trifluoromethylphenyl)methyl]phosphonate (**1**) and AgBF_4_ led to crystals of silver-*α*-aminophosphonate adducts with various compositions and structures (Figure 1). Infrared and mass spectrometry analysis carried out on the crystals revealed that the arrangement of the formed complexes depended exclusively on the metal-to-ligand ratio (Ag/**1**) used for the synthesis. Moreover, the IR spectrum of complex **2**, unlike complex **3**, displayed two weak bands at 724 and 751 cm^−1^ and five broad bands in the range of 925–1067 cm^−1^ consistent with a coordinated BF_4_ anion to a silver atom (see Appendix A) [59].

Despite partial decomposition of complex **2** during the ESI-TOF analysis, cations with two silver atoms and a BF_4_ counter anion could be observed at *m*/*z* = 1666.18 [M + **1** − BF_4_]^+^ and 2121.30 [M + (**1**)_2_ − BF_4_]^+^. On the other hand, mass spectrum analysis of complex **3** displayed cations corresponding to *m*/*z* = 562.03 [M − (**1**)_3_ − BF_4_]^+^, 1019.15 [M − (**1**)_2_ − BF_4_]^+^, 1474.27 [M − **1** − BF_4_]^+^, and 1929.40 [M − BF_4_]^+^ (Figure 2). Indeed, two different silver complexes, **2** ([Ag_2_(**1**)_2_(BF_4_)_2_]) and **3** ([Ag(**1**)_4_]BF_4_), were obtained starting from Ag/**1** ratios of 1/1 and 1/4, in 42 and 28% yields, respectively (Figure 1).

### 2.2. Single-Crystal X-ray Diffraction Studies

The binuclear complex **2** crystallized in the monoclinic asymmetric space group *P*2_1_/c and contained two silver atoms, two distinct enantiomeric *α*-aminophosphonate **1** (the two C9 atoms have an (*R*)- and an (*S*)-configuration), and two BF_4_ anions. The CF_3_ (F1, F2, and F3) moiety was disordered over two positions with a ratio of 0.6/0.4 (Figure 3 and Table 1). The silver atoms, which adopted {AgN_2_F} planar trigonal geometries, were simultaneously bridged by two oxadiazole rings of two ligands with Ag1-N1 and Ag1-N2 bonds of 2.260(2) and 2.215(2) Å, respectively. The coordination spheres were completed with the coordination of the counter anion BF_4_ through a fluorine atom [23] (Ag-F4 2.652(2) Å; sum of van der Waals radii 3.27 Å [60]). The counter anion was also maintained by a hydrogen bond with the amine (NH3···F5 2.117 Å). The distance between the two silver ions was 3.417 Å. This distance was slightly smaller than the sum of the van der Waals radii of two silver ions (3.44 Å [60]), suggesting the occurrence of very weak argentophilic interactions [61]. The oxadiazole rings were planar with the phenyl substituents (dihedral angle of 3.03°) and angled with respect to the 4-trifluoromethylphenyl ring (dihedral angle of 44.21°).

The mononuclear complex **3** crystallized in the non-usual cubic space group *Fd-3*c. The silver atom bound to four distinct enantiomeric *α*-aminophosphonates (the four C9 atoms had twice (*R*)- and twice (*S*)-configuration). The CF_3_ (F1, F2, and F3) and ethyl (C10 and C11) moieties were disordered over two positions with ratios of 0.5/0.5 and 0.57/0.43, respectively (Figure 4 and Table 1). The silver atoms adopted {AgN_4_} slightly distorted tetrahedral geometries with Ag1-N1 bonds of 2.289(4) Å and N1-Ag1-N1 angles of 105.56(10) or 117.6(2)°. The dihedral angles between the oxadiazole and the phenyl or the 4-trifluoromethylphenyl rings were 12.43 and 53.89°, respectively.

The study revealed the presence in the solid state, for each silver complex, of eight hydrogen bonds involving their N-H and P=O moieties [62] (H···O 1.877 Å) supramolecularly interlinked to each complex with four neighbors around the three crystallographic axes. In the crystal lattice (118,085(4) Å^3^), the 48 [Ag(**1**)_4_] cations generated a three-dimensional microporous supramolecular network, which contained a void between the inorganic molecules (18,208 Å^3^). At the end of the crystal structure determination, residual electron density peaks (4825 e-) were found for which no satisfactory disorder model could be obtained. The disordered species were a mixture of counter anion BF_4_ (1968 e-) and solvent molecules (2857 e-), occupying the space left vacant by the silver complexes [31]. The SQUEEZE program was used to account for the disorder. The contribution of these species was removed from the final structure factor calculations (see Appendix A).

It is interesting to note that in both silver complexes, the potentially coordinating group P(O)(OEt)_2_ was not involved in a coordination with the metal centers.

### 2.3. In Vitro Effects Assessment on MCF-7 and PANC-1 Cell Lines

Preliminary in vivo toxicity studies carried out on brain of mice demonstrated that the monomeric silver complex **3** is not neurotoxic contrary to the dimeric complex **2** [63]. Consequently, the antitumor potential of the *α*-aminophosphonate **1** and the tetrahedral silver complex **3** were evaluated against MCF-7 and PANC-1 cells by a one-dose assay. The 3-(4,5-dimethythiazol-2-yl)-2,5-diphenyl tetrazolium bromide (MTT) assays were performed to investigate the dose-dependent silver compounds’ effect on cell viability. For this purpose, compounds were added on cells at an increasing concentration (3.125–200 μM) during 24 and 48 h and IC_50_ values were calculated (Table 2). After 24 h of incubation, the results indicated that *α*-aminophosphonate **1** was about twice less cytotoxic than the silver complex **3**, and IC_50_ values of 22.8 ± 0.8 and 11.2 ± 0.4 μM were obtained when the MCF-7 cell line was treated with **1** and **3**, respectively (Figure 5). Both compounds had a dose-dependent effect and, when exposed to the PANC-1 cell line, they showed lower cytotoxicity with IC_50_ values of 35.0 ± 1.0 and 18.3 ± 1.2 μM for **1** and **3**, respectively (Figure 6).

The two drugs **1** and **3** had a time-dependent effect; the cytotoxicities could be improved by extending the incubation time to 48 h. Thus, with the silver complex **3**, IC_50_ values of 8.3 ± 1.0 and 14.4 ± 0.6 μM were observed on MCF-7 and PANC-1 cell lines, respectively. The IC_50_ values are slightly higher than those previously reported in the literature for cisplatin, 5.7 ± 0.1 (incubation for 24 h) and 14.4 ± 1.1 (incubation for 48 h) for MCF-7 [64] and PANC-1 [65] cell lines, respectively.

Comparison with other organometallic silver complexes tested on MCF-7 and PANC-1 cell lines, such as silver(I) complexes in which the metal is coordinated to carbon [6,66], phosphorus [67,68], or nitrogen [69] atoms, showed that *α*-aminophosphonate **1** and complex **3** exhibited comparable effects (Figure 7).

### 2.4. Molecular Docking Studies

Molecular docking is a frequently used method in detailed bioactivity analysis of molecules [70]. It is possible to make qualitative and quantitative evaluations of the interactions between target macromolecule crystals and candidate bioactive molecules [71]. While the interaction areas qualitatively indicate how the ligand and the biological macromolecule will interact, the relative value of the binding can be considered as a quantitative measurement of the affinity. High ligand binding affinity means both stronger interaction between the ligand and target molecule and measurement of the ligand concentration required for inhibition [72].

Boda et al. synthesized some new isatin-1,2,4-oxadiazole derivative molecules and analyzed the activity of these molecules against some cancer cell lines including MCF-7. The authors analyzed the interactions between these molecules and the VGEFR-2 crystal structure and recorded a binding energy of −10.85 kcal/mol [73]. Serdaroğlu et al. reported the in vitro cytotoxic activities of silver complexes coordinated to *N*-heterocyclic carbene ligands. The study showed that the organometallic molecules could be efficient anti-cancer agents with a binding affinity of −7.59 kcal/mol [74]. Although there are many studies of molecular docking, which analyzed the anticancer activities of aminophosphonate derivatives and their metal complexes, a finite number of them have focused on the interactions with VGEFR-2 for a detailed analysis of anticancer activity [75,76,77,78,79].

In our work, the *α*-aminophosphonate **1** was optimized with the ORCA package program [80,81] and the most stable structure interacted with the VEGFR-2 crystal structure [82]. Some interactions for the ligand were detected: (i) four H-bonds between the ligand and Asp1044, Cys1043, Glu883, and Lys866 with lengths of 2.19, 3.66, 2.91, and 3.00 Å, respectively; (ii) π-alkyl interactions with Ala864, Val897, Val914, Cys917, and Leu1033; (iii) van der Waals interactions; (iv) halogenic interaction of the CF_3_ with Arg 1030. The total energy of all these interactions was −7.58 kcal/mol. The binding sites of the ligand and the silver complex **3** were different due to the larger steric hindrance of the complex. Complex **3** carried out fewer interactions but they were stronger than **1**, and the most important interactions were H-bonds with Asp1044, Arg1078, and Arg1025 with lengths of 2.75, 3.20, and 3.07 Å, respectively. Additional van der Waals and alkyl interactions contributed to the total binding between the silver complex and the target molecule. The total value of these interactions was calculated at −7.82 kcal/mol (Figure 8). The results show that the silver complex **3** will bind more effectively than the *α*-aminophos-phonate **1**, and will, therefore, be a more effective inhibitor. These results are also in agreement with the experimental anticancer results vide supra.

Molecular docking is also a frequently used method for analyzing the interaction of molecular pharmaceuticals with DNA [83]. The B-DNA dodecamer (1bna) crystal structure was used to determine the interaction modes of molecules with DNA [84]. Both molecules (**1** and **3**) interact with approximately the same regions of DNA. While the *α*-aminophosphonate **1** interacted with the region consisting of Thy8, Cyt9, Gua10, Cyt11, and Gua12, the silver complex **3** additionally interacted with Ade6 and Thy7. The final interaction energies were determined as −6.9 and −7.86 kcal/mol for **1** and **3**, respectively (Figure 9). 

## 3. Materials and Methods

All manipulations were carried out under dry argon. Solvents were dried by conventional methods and were distilled immediately before use. Routine ^1^H, ^31^P{^1^H}, and ^19^F{^1^H} spectra were recorded with Bruker FT instruments (AC 300 and 500). ^1^H NMR spectra were referenced to residual protonated solvents (*δ* = 7.26 ppm for CDCl_3_). ^31^P and ^19^F NMR spectroscopic data are given relative to external H_3_PO_4_ and CCl_3_F, respectively. Chemical shifts and coupling constants are reported in ppm and Hz, respectively. Infrared spectra were recorded with a Bruker FT-IR Alpha-P spectrometer. The optical density was measured at 570 nm with an ELISA Microplate Reader spectrophotometer. Elemental analyses were carried out by the Service de Microanalyse, Institut de Chimie, Université de Strasbourg. Diethyl[(5-phenyl-1,3,4-oxadiazol-2-ylamino)(4-trifluoro-methylphenyl)methyl]phosphonate (**1**) was prepared by a literature procedure [85].

### 3.1. General Procedure for the Synthesis of the Silver(I) Complexes

A solution of Ag(BF_4_) (0.019 g, 0.10 mmol) in THF (2 mL) was added to a solution of diethyl[(5-phenyl-1,3,4-oxadiazol-2-ylamino)(4-trifluoromethylphenyl)methyl]phosphonate (**1**, 0.10 or 0.4 mmol for the synthesis of **2** and **3**, respectively) in THF (10 mL). The reaction mixture was stirred in the dark for 16 h at room temperature. The solvent was removed under reduced pressure. Crystals were grown from a slow diffusion of hexane into a solution of the crude product, dissolved into the minimal amount of THF. The colorless crystals of the silver complex were collected and dried under vacuum. 

#### 3.1.1. Bis{diethyl[(5-phenyl-1,3,4-oxadiazol-2-yl-κ*N*^3^:κ*N*^4^-amino)(4-trifluoromethyl-phenyl)methyl]phosphonate-(tetrafluoroborato-κ*F*)}-di-silver(I) (**2**)

Yield 42%. ^1^H NMR (300 MHz, CDCl_3_): *δ* = 7.85 (dd, 4H, arom. CH of CF_3_C_6_H_4_, ^3^*J*_HH_ = 6.9 Hz, ^4^*J*_HH_ = 1.8 Hz), 7.71–7.62 (m, 8H, arom. CH), 7.49–7.42 (m, 6H, arom. CH), 5.28 (d, 2H, CHP, ^2^*J*_PH_ = 22.5 Hz), 4.24–4.11 (m, 4H, OC*H*_2_CH_3_), 4.03–3.81 (m, 4H, OC*H*_2_CH_3_), 1.32 (t, 6H, OCH_2_C*H*_3_, ^3^*J*_HH_ = 7.2 Hz), 1.15 (t, 6H, OCH_2_C*H*_3_, ^3^*J*_HH_ = 7.1 Hz) ppm, in the spectrum, the two NH protons are not observed; ^31^P{^1^H} NMR (121 MHz, CDCl_3_): *δ* = 19.1 (q, P(O), ^7^*J*_PF_ = 2.0 Hz) ppm. ^19^F{^1^H} NMR (282 MHz, CDCl_3_): *δ* = −62.72 (d, CF_3_, ^7^*J*_PF_ = 2.2 Hz), −147.92 (brs, BF_4_) ppm. IR: *ν* = 724, 751, 925, 954, 1017, 1031, 1067 (F_3_BF-Ag), 1658 cm^−1^ (C=N). MS (ESI-TOF): decomposition of the complex occurred during the analysis, although some cations containing two silver complex could be observed at *m*/*z* = 1666.18 [M + **1** − BF_4_]^+^ and 2121.30 [M + (**1**)_2_ − BF_4_]^+^ (expected isotopic profile). Elemental analysis calcd. (%) for C_40_H_42_Ag_2_B_2_F_14_N_6_O_8_P_2_ (1300.08): C 36.95, H 3.26, N 6.46; found C 35.85, H 3.14, N 6.27.

#### 3.1.2. Tetrakis-{diethyl[(5-phenyl-1,3,4-oxadiazol-2-yl-κ*N*^3^-amino)(4-trifluoromethyl-phenyl)methyl]phosphonate}silver(I) Tetrafluoroborate (**3**)

Yield 28%. ^1^H NMR (300 MHz, CDCl_3_): *δ* = 7.86 (dd, 8H, arom. CH of CF_3_C_6_H_4_, ^3^*J*_HH_ = 7.8 Hz, ^4^*J*_HH_ = 1.2 Hz), 7.74–7.62 (m, 16H, arom. CH), 7.54–7.44 (m, 12H, arom. CH), 5.34 (d, 4H, CHP, ^2^*J*_PH_ = 21.9 Hz), 4.24–4.14 (m, 8H, OC*H*_2_CH_3_), 4.08–3.90 (m, 8H, OC*H*_2_CH_3_), 1.32 (t, 12H, OCH_2_C*H*_3_, ^3^*J*_HH_ = 7.0 Hz), 1.20 (t, 12H, OCH_2_C*H*_3_, ^3^*J*_HH_ = 7.0 Hz) ppm; in the spectrum, the four NH protons are not observed; ^31^P{^1^H} NMR (121 MHz, CDCl_3_): *δ* = 17.9 (q, P(O), ^7^*J*_PF_ = 2.2 Hz) ppm. ^19^F{^1^H} NMR (282 MHz, CDCl_3_): *δ* = −62.77 (d, CF_3_, ^7^*J*_PF_ = 2.2 Hz), −148.47 (s, BF_4_) ppm. IR: *ν* = 1633 cm^−1^ (C=N). MS (ESI-TOF): *m*/*z* = 562.03 [M − (**1**)_3_ − BF_4_]^+^, 1019.15 [M − (**1**)_2_ − BF_4_]^+^, 1474.27 [M − **1** − BF_4_]^+^ and 1929.40 [M − BF_4_]^+^ (expected isotopic profile). Elemental analysis calcd. (%) for C_80_H_84_AgBF_16_N_12_O_16_P_4_ (2016.14): C 47.66, H 4.20, N 8.34; found C 47.54, H 4.33, N 8.21.

### 3.2. X-Crystal Structure Analysis

Single crystals suitable for X-ray analysis were obtained by the slow diffusion of hexane into a THF solution of the complex. The samples were studied on a Bruker PHOTON-III CPAD and Bruker APEX-II CD for complexes **2** and **3**, respectively, using Mo-*K*_α_ radiation (*λ* = 0.71073 Å) at T = 120(2) K. The structures were solved with SHELXT-2014/5 [86] (complex **2**) or SHELXS-97 [87] (complex **3**), which revealed the non-hydrogen atoms of the molecule. After anisotropic refinement, all of the hydrogen atoms were found with a Fourier difference map. The structure was refined with SHELXL-2014/7 [88] by the full-matrix least-square techniques (use of *F* square magnitude; *x*, *y*, *z*, *β*_ij_ for C, Ag, B, F, N, O, and P atoms; *x*, *y*, *z* in riding mode for H atoms).

### 3.3. Cell Culture and Cell Viability Assay

The experiments were performed on MCF-7 (human breast cancer, HTB-22) and PANC-1 (human pancreatic cancer, CRL-1469) cell lines. The cell lines obtained from the American Type Culture Collection and were cultivated on 10% fetal bovine serum (FBS), 1% L-glutamine, and 100 IU/mL of penicillin and 10 mg/mL of streptomycin containing high-glucose Dulbecco’s modified Eagle’s medium (DMEM) (Gibco™, Sigma). Cells were cultivated in a humidified atmosphere with 5% CO_2_ at 37 °C.

The cytotoxic activity of compound **1** and silver complex **3** on cells was evaluated using the 3-(4,5-dimethythiazol-2-yl)-2,5-diphenyl tetrazolium bromide (MTT) method as previously described by Mosmann [89] with slight modifications. Cells (7000 per 200 μL) in medium were seeded into 96-well microplates. Compound **1** or **3** was added to the wells by serial dilution (200 μM, 100 μM, 50 μM, 25 μM, 12.5 μM, 6.25 μM, and 3.125 μM) and the cells were incubated at 37 °C in a 5% CO_2_ incubator for 24 h and 48 h. At the end of the incubation time, the MTT reagent (20 μL) was added and the cells were incubated for a further 4 h. The formed formazan crystals were dissolved in 100 μL of DMSO and the absorptions were measured at 570 nm with an ELISA Microplate Reader. Experiments were performed five times and the IC_50_ values, which corresponds to the concentration required for 50% inhibition of cell viability, were calculated by Graphpad prism software (version 8.2.1, San Diego, CA, USA). Results were expressed as percentage of viable cells relative to the negative control (untreated cells). 

### 3.4. Molecular Docking Calculation Methods

The molecular docking procedures were performed with AutoDock 4.2 [90] against VGEFR-2 and B-DNA dodecamer crystal structures [82,84]. All the structures were downloaded from the RCSB protein data bank (https://www.rcsb.org/, accessed on 20 October 2022) (PDB ID: 1ywn for VGEFR-2; 1bna for B-DNA dodecamer, accessed on 16 August 2022). The maximum torsion number with the fewest atoms was set for the ligand molecules. Kollman charges were regarded, and only polar hydrogens were employed in all target crystal structures. Water molecules in the targets were extracted as well as Gasteiger charges, and randomized starting positions were utilized during the processes. Lamarckian genetic algorithms were employed with 150 genetic algorithm populations for 10 runs [91]. The standard program defaults were used during the running. All the illustrations were performed with Discovery Studio 4.1.0. (BIOVIA Corp, San Diego, CA, USA).

## 4. Conclusions

A simple modification of the silver salt to ligand, bearing an oxadiazole ring, to silver salt ratio drastically modified the coordination sphere of silver(I) cations, leading either to a trigonal dimeric or to a tetrahedral monomeric silver complex. The α-aminophosphonate and the monomeric silver complex were evaluated in vitro against MCF-7 and PANC-1 cell lines by a one-dose assay. The presence of a silver atom had a beneficial effect on the cytotoxic activities, and approximately twofold lower half-maximal inhibitory concentration (IC50) values were observed when the organometallic complex was employed compared to the organic drug. The silver complex is promising as a drug candidate for breast cancer and the pancreatic duct and future studies will be devoted to investigate its mode of action along with its in vivo effectiveness and safety for clinical application. The anticancer activities of the molecules were also confirmed and detailed by molecular docking methods. The results show that the silver complex could be a more effective anticancer agent than the ligand and this in silico result is in accordance with the experimental ones.

## Data Availability

The data presented in this study are available on request from the corresponding author.

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
