# Peer review of "Silver(I) Complexes Based on Oxadiazole-Functionalized α-Aminophosphonate: Synthesis, Structural Study, and Biological Activities"

_molecules, 2022, doi:10.3390/molecules27238131_

Round 1

Reviewer 1 Report

The paper deals with the synthesis and study of two silver complexes of some oxadiazole derivatices. The complexes are studies by different techniques as IR, MS and x.ray studies showing two very interesting complexes wich are studies agains to cancer cell lines. The paper mirits publication with minor changes:

-When firstly described complexes (line 120) a reference to scheme 1 must be included, and the fig 1 could  be send to supplementary material as bigger figures.

In line 202 the sentence the the complex 2 is not toxic should be implemented with some data that shows this behaviour and explain why the complex 2 is not in the cancer study.

In the Materials and Methods   the 19F{1H} of two complexes (lines 321 and 335) must be d, doublets, instead q

Author Response

We thank this referee for his/her comments and careful reading of the manuscript.

* “When firstly described complexes (line 120) a reference to scheme 1 must be included, and the fig 1 could be send to supplementary material as bigger figures.”

The reference to Scheme 1 was added and the figure was moved to supplementary materials as bigger figures.

* “In line 202 the sentence the complex 2 is not toxic should be implemented with some data that shows this behavior and explain why the complex 2 is not in the cancer study.”

The following comment was added “Preliminary in vivo toxicity studies carried out on brain of mice demonstrated that the monomeric silver complex 3 is not neurotoxic contrary to the dimeric complex 2 [63].”.

* “In the Materials and Methods the 19F{1H} of two complexes (lines 321 and 335) must be d, doublets, instead q.”

The two mistakes were corrected.

Reviewer 2 Report

The manuscript presents an interesting topic for medicinal chemistry and is well presented, however the enzyme assay on VEGFR is highly recommend for both compound 1 and 3 against a reference inhibitor to support the molecular modeling study.

Author Response

We thank the referee for his/her insightful remark.

* “The enzyme assay on VEGFR is highly recommend for both compound 1 and 3 against a reference inhibitor to support the molecular modeling study.

In this preliminary work, we only analyzed the cytotoxic activity of the compounds 1 and 3 and tried to provide evidences of probable and favorable interactions between our compounds and VEGFR-2 through theoretical molecular docking. Unfortunately, it is impossible for us to obtain in a reasonable time the materials necessary to carry out enzyme assay on VEGFR-2. This work will be the subject of future investigations, which will allow us a better understanding of the mode of action of compounds 1 and 3.